# Dynamics of the Inbreeding Coefficient and Homozygosity in Thoroughbred Horses in Russia

**DOI:** 10.3390/ani10071217

**Published:** 2020-07-17

**Authors:** Valery Kalashnikov, Lyudmila Khrabrova, Nina Blohina, Alexander Zaitcev, Tatyana Kalashnikova

**Affiliations:** The All-Russian Research Institute for Horse Breeding, Ryazan region, 391105 Rybnoe, Russia; vniik08@mail.ru (V.K.); nbloh16@yandex.ru (N.B.); amzaitceff@mail.ru (A.Z.)

**Keywords:** genetic parameters, genetic diversity, inbreeding, horse, Thoroughbred

## Abstract

**Simple Summary:**

The Thoroughbred horse breed provides a unique model for studying the theoretical bases and practical methods of selection in horse breeding. To assess the diversity in the modern population of Russian Thoroughbred horses, we used classic parameters, such as Wright’s inbreeding coefficient, the heterozygosity level, and population inbreeding level, over the period 1990–2018. The average inbreedin***g*** coefficient of the Thoroughbred horses was 0.83% and slowly increased over the past three decades. The individual observed heterozygosity levels of the tested horses varied within 23.1–100%, and the population’s parameters of observed heterozygosity tended to decrease. The relationship between the level of inbreeding and the rate of homozygosity was not significant (*r* = 0.022; *p* > 0.05).

**Abstract:**

The Thoroughbred (TB) horse has hugely impacted the development of horse breeding around the world. This breed has unique genetic qualities due to having had a closed studbook for approximately 300 years. In Russia, TBs have been bred since the second half of the 18th century. Here, we analyzed the genetic diversity and the inbreeding level in TB horses (*n* = 9680) for the period from 1990 to 2018 using polymorphisms of 17 microsatellite loci. We found that the genetic structure of the TB breed in Russia is represented by 100 alleles of panel STR (short tandem repeat) loci and has been stable for the past three decades. The conducted monitoring revealed a slight increase in the Wright’s inbreeding coefficient in all age and sex groups of TB horses (stallions, broodmares, and foals) from 0.68% to 0.90%, which was followed by a decrease in the degree of heterozygosity, Ho, from 68.5% to 67.6%. The Spearman’s rank correlation coefficient between the level of inbreeding and the degree of homozygosity was estimated (*r* = 0.022; *p* > 0.05). The obtained data on the DNA genotypes of horses of different breeds provide a unique base for the evaluation of genetic variability and the control of genetic variability of horses in selection programs.

## 1. Introduction

The Thoroughbred (TB) breed of horses, created in England in the 17–18th centuries, was perfected using purebred breeding for many generations. The breed was formed as a result of crossing local mares with stallions of Eastern origin. The breed has three foundation stallions: the Byerley Turk, the Darley Arabian, and the Godolphin Arabian [1]. The gene pool of this breed was formed by several hundred horses recorded in Volume 1 of the General Studbook (1793), where data on their origin starting in 1660 were recorded [2]. TB horses showed good racing ability, so they were bred in many European countries. In Russia, the first volume of the Studbook, which included 654 TBs, was published in 1836 [3].

The long-term intensive selection of horses for racing performance and the wide use of the best stallions in breeding led to the accumulation of well-known names in their pedigrees. According to Cunningham et al. [4], TB horses have a narrow genetic base, with 78% of the alleles in the current population being from 30 progenitors, whereas one stallion, the Godolphin Arabian, is responsible for 95% of the paternal lines. This may have led to an increase in the inbreeding level and the homozygosity rate of the genomes of TB horses [5,6].

Mating among relatives is one of the most important breeding methods used to consolidate the hereditary qualities of animals. However, the negative effects of related mating are well known, and are expressed in an increase in the homozygosity rate of animals and inbred depression [7,8,9]. The importance of preserving the sources of genetic diversity in breeds necessitates the development of optimal parameters for the use of inbreeding in horse breeding programs, including optimal breeding policies [10] and the control of the homozygosity of animals and populations [11].

The practice of related breeding shows that animal breeds differ in the rates at which homozygosity increases (that is, the inbreeding rate). A particular inbreeding level exists for each breed, and exceeding this level leads to inbred depression. According to Pern [9], inbred depression manifests at an inbreeding rate of over 3.12% for TB horses in Russia.

The measurement of the actual homozygosity level at the individual and population levels only became possible at the end of the 20th century due to the development of methods for studying genes and DNA polymorphisms. As a result of the widespread use of microsatellite loci in the control of animal origins, this type of highly polymorphic genetic marker has been successfully used to study the genetic features of breeds and their phylogenetic relationships [12,13,14,15]. The characteristics of the genetic parameters of populations, among which the heterozygosity level is one of the most important, allow us to assess the impact of the breeding system on the level of inbreeding [16,17,18,19,20].

Assessment of the inbreeding and homozygosity levels in TB horses is particularly relevant, as the breed has been improved only through purebred breeding for more than three centuries. Monitoring the state of genetic diversity in the Russian population of TB horses in the period from 1980 to 2005 revealed the tendency towards a certain decrease in the level of polymorphisms and the rate of heterozygosity, especially in the group of stallions [21].

Due to the import of a relatively large number of TB horses from different countries around the world to Russia in recent decades, changes have occurred in the genealogical structure of the Russian horse population, which has led to the descendants of Northern Dancer (30.9%) and Nasrullah (16.7%) dominating in the linear structure of the breed. These significant changes could have a certain impact on the level of the genetic diversity of the population [22].

The purpose of our research was to assess the genetic diversity and the level of inbreeding in the TB breed in Russia over the period 1990–2018.

## 2. Materials and Methods

All data for this study were obtained in the Laboratory of Genetics of the All-Russian Research Institute for Horse Breeding, certified by ISAG (International Society for Animal Genetics) based on the Horse Comparison Test (HCT) results.

### 2.1. Sample Collection, Genomic DNA Isolation, and Inbreeding Recording

The materials for the research included the database of the DNA typing of 9680 TB horses registered in the Russian Thoroughbred Stud Book for the period of 1996–2018, which includes approximately 98% of the horses bred in Russia. The studied population included 957 stallions, 3194 broodmares, and their offspring. Genomic DNA samples were prepared from hair follicles or blood samples of TB horses by standard procedure using the “Extra Gene^TM^ DNA Prep 200” and “Diatom^TM^ DNA Prep 200” kits (Laboratory Isogene, Moscow, Russia) in the Laboratory of Genetics of the All-Russian Research Institute for Horse Breeding. The individual inbreeding coefficient of horses was determined using Wright’s formula [8], considering five generations of ancestors of the pedigree, using the base of the information-searching system for managing the selection process in Russian horse breeding “ISS Kony 3” (ARRIH, Divovo, Russia).

The dynamics of the changes in the level of inbreeding in groups of stallion, mares, and foals was analyzed considering three periods (1990–1999, 2000–2009, and 2010–2018), corresponding to the generation change interval in horse breeding.

### 2.2. PCR Amplification and Genotyping Microsatellite Loci

Isolated DNA probes were amplified on a Thermocycler 2730 (Applied Biosystems, Foster City, CA, USA) using 17-plex kit primers for horses (StockMarks^®^, Applied Biosystems, Foster City, CA, USA). The kit includes 17 loci (AHT4, AHT5, ASB2, ASB17, ASB23, CA425, HMS1, HMS2, HMS3, HMS6, HMS7, HTG4, HTG6, HTG7, HTG10, LEX3, and VHL20) recommended for Thoroughbred parentage testing by the Standardization Committee of the ISAG [23]. The PCR amplification program consisted of 10 min at 95 °C, followed by 32 cycles at 95 °C for 30 s, 60 °C for 30 s, and 72 °C for 60 s. After the final cycle, an extension step of 72 °C for 5 min was conducted, and then the samples were cooled to 4 °C. The purified PCR products of 17 panel microsatellites were analyzed on the automated DNA analyzer ABI 3130 (Applied Biosystems, Foster City, CA, USA). The size of the alleles was determined using the standard GeneScan-LIZ 500 and Sequencing Analysis 4.0 software (Applied Biosystems, Foster City, CA, USA) according to the manufacturer’s instructions. We also used the data of the control DNA probes and the Horse Comparison Test (HCT) results from ISAG. The characteristics of the 17 microsatellite loci, used in our study, are presented in Table 1. 

### 2.3. Statistical Analyses

The genetic diversity in groups of TB horses was evaluated using basic parameters, including the total number of allele variants (*N_a_*), the effective number of alleles (*A_e_*), the number of alleles per loci (*MNA*), the observed (*H_o_*) and expected (*H_e_*) heterozygosity, and the Hardy–Weinberg equilibrium (HWE). Intra-breed inbreeding was estimated using F-statistics [24], using the FSTAT 1.2. program (Goudet, Lausanne, Switzerland). The relationship of the homozygosity level of microsatellite loci and inbreeding coefficient was analyzed with the Spearman rank correlation coefficient using Statistica 12 ver.10 software (StatSoft, Tulsa, OK, USA). 

## 3. Results

As a result of the genotyping of TB horses, a total of 100 alleles were identified using 17 panel STR loci, among which the frequency of the most common 70 variants exceeded 0.05. In the periods under study, the number of rare alleles in the STR loci in tested horses varied slightly in the range of 27 to 29 and was the highest in the second period (2000–2009), which was associated with the intensive import of TB horses from Europe and the USA. During this period, the allele pool of the domestic population was supplemented with three new rare alleles, ASB17H, HMS1L, and LEX3I, due to the genotypes of the imported mares. However, in the next generation, in horses born in 2010–2018, the HMS1L allele was lost. The total number of alleles on 17 STR loci was only 98 in the modern Russian population of TB horses (third period). The breed stably maintained its genetic structure despite continued imports of horses from other countries. 

The intensive system of breeding TB horses for racing performance was not found to contradict the fairly stable state of the current genetic structure of the breed, which changed insignificantly over the studied 30 years. We found slight fluctuations in the frequency of occurrence of both typical and rare alleles, which invariably retained their status. During the analyzed period, we noted a trend of an increase in the concentration of the major allele HMS2L (0.648–0.749) (data not shown), found in the genotypes of the best stallions. Earlier analysis showed that this locus may be associated with genes that influence the distance abilities of Thoroughbred horses [25].

A comparative analysis of the main genetic and population indicators by period showed that the studied characteristics changed insignificantly over three generations from 1990 to 2018: *Na* varied between 5.71 and 5.77 from 1990 to 2018, with a maximum of 5.82 reached in 2000–2009; *Ae* varied between 3.508 and 3.411; and *Ho* varied between 0.682 and 0.674. The highest level of genetic diversity of the Russian population of TB horses was observed in the late 1990s (first period), when the largest number of breeding broodmares in the country was registered with studs and on breeding farms. Increased imports of TB horses in the second period slightly increased the variability of rare alleles but did not significantly change the genetic structure of the breed (Table 2). The monitoring revealed a trend of a slight decrease in the genetic diversity in the population of TB horses in Russia in recent decades for all basic indicators (*Na*, *Ae*, and *Ho*; Table 3).

In the first and third periods, the number of heterozygous genotypes corresponded to the theoretically expected values, which was confirmed by negative Fis values. Violation of the gene balance in the breed (Fis = 0.002) was registered only in the second period, and, as the analysis of genotypes showed, this occurred due to the gene drift that accompanied the increased import of TB horses from other countries. 

Due to the imported mares, the allele pool of the Russian population of the TB breed was supplemented with the new rare alleles ASB17H, HMS1L, and LEX3I. Mares born in other countries and imported in 1995–2009 were characterized by a significantly higher frequency of the alleles ANT5K, ASB2Q, HMS7J, and HTG4 K (*p* < 0.001) (data not shown).

### Association between the Inbreeding Coefficient and Level of the Heterozygosity 

A comparison of the average values of the inbreeding coefficient in TB horses of different ages and sex groups showed that stallions, broodmares, and foals in the first period were characterized by the lowest degree of inbreeding (Table 4). The maximum inbreeding rate in the pedigrees of TB horses was 13.28%, and this was observed only in one of the stallions. Stud broodmares had the highest inbreeding rate of 12.70%.

The monitoring revealed a trend of a gradual increase in the coefficient of inbreeding in all age and sex groups of TB horses (stallions, broodmares, and foals), which averaged 0.68–0.81–0.90% over the periods.

In the 1990s, the share of outbred animals in the analyzed groups of horses ranged from 43.37% to 47.76%, while the highest rate was observed among the stallions. Inbreeding coefficients of 3.1% and higher were only found in 4% of stallions, 7.4% of mares, and 7.1% of foals.

In the first decade of the 21st century, with the background of an increase in the inbreeding coefficient and an increase in the degree of homozygosity of STR loci, the percentage of outbred horses in groups decreased to 37.7–41.4%. However, the share of horses with inbreeding above 3.1% increased slightly to 6.1–7.2% (data not shown).

In the third period (2009–2018), the tendency of a reduction in the number of horses with outbred pedigrees in the livestock structure remained. As a result, the share of outbred young animals decreased to 29.7%, and there was a slight increase in the number of horses (36.5–42.7%) with a low inbreeding rate, at the up to 1% level. Inbreeding at the level of 3.1% and higher occurred in 5.6–8.1% of horses, while the maximum indicator was identified in the group of broodmares (Figure 1).

The data presented in Table 5 characterize the distribution of TB horses by the degree of homozygosity and the inbreeding level (Fx). The majority of TB horses in the three studied decades were obtained by outbreeding (37.96%) or remote and moderate inbreeding (35.96%). Regardless of the value of the inbreeding coefficient, the modal class of the degree of homozygosity was 25.3–31.5%.

The lowest level of homozygosity was found in the group of stallions, but, in general, the differences between the compared groups of horses were slight (Table 6). Stallions had the lowest coefficient (Fx = 0.747) on average. The group of foals had slightly higher rates of homozygosity (32.4%) and inbreeding coefficients (Fx = 0.838).

Changes in the level of homozygosity in different groups of TB horses were insignificant, whereas a slight increase in this indicator was observed in stallions and broodmares with an inbreeding coefficient of 3.1–4.0%.

The Spearman rank correlation coefficient between the inbreeding level and degree of homozygosity of the Thoroughbred horses was close to zero (*r* = 0.022; *p* > 0.05). A statistically significant value of this indicator was found only in the group of broodmares (*r* = 0.030, *p* = 0.037) (data not shown).

## 4. Discussion

The results of genotyping of 9680 Thoroughbred horses, registered in the Russian Stud Book, showed that the studied group had a typical spectrum of alleles of microsatellite loci for the breed and was characterized by a high degree of consolidation. Only insignificant differences in the frequency of rare alleles were found in the TB populations of foreign countries. For example, a small population of Thoroughbred horses from Bosnia and Herzegovina [26] had higher variability of the same STR loci (in total 103 alleles). The results of the comparative analysis of the genetic structure of different TB populations [27,28] showed that the TB horses had their own pool of alleles that remained stable for generations. The modern Russian TB population was characterized by a fairly high level of genetic diversity (Ae = 3.41, Ho = 0.676) and lack of population inbreeding (Fst = −0.002).

Despite the expectations, the periodic import of TB horses from the United States and European countries had no significant impact on the population parameters of the genetic structure of TB horses in Russia. This led to an increase in the inbreeding coefficient and level of homozygosity. The comparative analysis of the genetic diversity in groups of horses born in Russia and in imported horses showed that domestic stallions and mares had higher heterozygosity levels [29], which was confirmed by the genetic parameters of the breed in the period of 1990–2000.

The history of TB horse breeding in a closed Stud Book for three centuries could result in a reduction of the genetic diversity of the TB horses. The relatively high degree of homozygosity in the horses of this breed was confirmed using microsatellite loci by other researchers [13,14,15,16]. Genomic analysis using the 670,000 genotyping array showed that the TB breed had the highest level of homozygosity [6]. The results of genome-wide analyses of TB horses using a SNP50 (Single nucleotide polymorphism) bead chip indicated that inbreeding in the TB breed had increased over the past 45 years (*r* = 0.24; *p* < 0.001) [5]. The trend of a certain increase in the level of inbreeding and in the degree of homozygosity of the TB horses in Russia generally reflected the global trend of the decreasing genetic diversity of this breed.

The studied population steadily maintained a certain level of heterozygosity (Ho = 0.685–0.676), both during the period of partial isolation (1990–1999) and during the intensive import of horses (2000–2009). This provides proof of the hypothesis of the existence of optimal heterozygosity of populations [30]. This can also help to maintain a stable ratio of the intra- and interpopulation components of genetic diversity in the population system [31]. Due to the large number of TB horses and extensive breeding in various countries, the breed retains a significant reserve of variability, which is confirmed by the genetic differences in its linear structure [32].

The inbreeding coefficient at the pedigree level does not adequately reflect the degree of homozygosity of an individual [7]; a more accurate method is the genomic assessment using an SNP kit [19,33]. In our studies, the level of homozygosity of outbred horses using 17 standard microsatellite loci varied in a wide range from 0% to 76%, and the absence of duplicate ancestors in the pedigree of a horse did not guarantee high heterozygosity. Similar results were obtained in the analysis of the relation of the degree of homozygosity microsatellite loci and the level of inbreeding in 1194 Orlov Trotter horses [18]. This demonstrated that the degree of homozygosity of the outbred Orlov Trotter varied within a wide range from 0% to up to 75%; the breed also had a low coefficient of correlation between the degree of homozygosity of the microsatellite loci and the inbreeding level (0.079).

The Wright’s inbreeding coefficient approximately reflected the degree of homozygosity of the animals; therefore, genome monitoring is important for further improvement of the Thoroughbred breed. We have planned further research using SNP chips on Russian TB horses to potentially provide interesting new findings, e.g., the possibility to detect subpopulation structures or selection signatures.

## 5. Conclusions

The study of the genetic structure of TB horses in Russia by 17 microsatellite loci demonstrated 100 alleles that were typical of the global allelofond of the breed and that these were stable for the past three decades. A comparative analysis of the main genetic and population indicators by period showed that over three generations, from 1990 to 2018, the studied characteristics changed slightly: *Na* (5.71–5.77), *Ae* (3.508–3.411), and *Ho* (0.685–0676). The highest level of genetic diversity of the Russian population of TB horses was observed in the late 1990s (first period). The monitoring revealed a trend of a steady increase in the inbreeding coefficient in all age and sex groups of Thoroughbred horses (stallions, broodmares, and foals), which averaged, over the periods, 0.68–0.81–0.90%. The increase in the degree of inbred horses was accompanied by a decrease in the degree of heterozygosity of STR loci from 68.5% to 67.6%, which generally reflected the reduced genetic diversity in this breed.

The trend of an increase in the level of inbreeding and the degree of homozygosity of TB horses in Russia generally reflects the global trend of reduced genetic diversity in this breed due to changes in the modern genealogical structure and the strict selection for racing performance, associated with a wider use of the descendants of outstanding stallions and broodmares in breeding.

## Figures and Tables

**Figure 1 animals-10-01217-f001:**
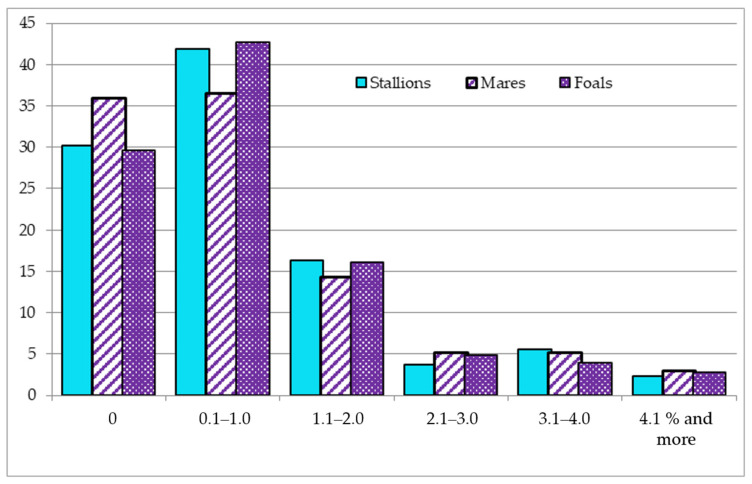
Distribution of Thoroughbred (TB) horses by inbreeding coefficient (Fx) in the period 2009–2018.

**Table 1 animals-10-01217-t001:** Information for the 17 microsatellite loci used in our study.

Locus	No. of Alleles	Chromosome Location	Amplicon Length (bp)	Primer Sequences (5′–3′)	Reference
AHT4	11	24	140–166	F: AACCGCCTGAGCAAGGAAGT R: CCCAGAGAGTTTACCCT	Binns et al., 1995 [5]
AHT5	9	8	126–147	F: ACGGACACATCCCTGCCTGC R: GCAGGCTAAGGGGGCTCAGC	Binns et al., 1995
ASB2	15	15	237–268	F: CCTTCCGTAGTTTAAGCTTCTG R: CACAACTGAGTTCTCTGATAGG	Breen et al., 1997
ASB17	19	2	104–116	F: GAGGGCGGTACCTTTGTACC R: ACCAGTCAGGATCTCCACCG	Breen et al., 1995
ASB23	14	3	176–212	F: GAGGTTTGTAATTGGAATG R: GAGAAGTCATTTTTAACACCT	Breen et al., 1995
CA425	11	28	224–247	F: AGCTGCCTCGTTAATTCA R: CTCATGTCCGCTTGTCTC	Der Valle A. et al., 1997
HMS1	7	15	166–178	F: CATCACTCTTCATGTCTGCTTGG R: TTGACATAAATGCTTATCCTATGGC	Guerin et al., 1994
HMS2	12	10	218–238/	F: ACGGTGGCAACTGCCAAGGAAG R: CTTGCAGTCGAATGTGTATTAAATG	Guerin et al., 1994
HMS3	10	9	146–170	F: CCAACTCTTTGTCACATAACAAGA R: CCATCCTCACTTTTTCACTTTGTT	Guerin et al., 1994
HMS6	7	4	154–170	F: GAAGCTGCCAGTATTCAACCATTG R: CTCCATCTTGTGAAGTGTAACTCA	Guerin et al., 1994
HMS7	9	1	167–186	F: CAGGAAACTCATGTTGATACCATC R: GTTGTTGAAACATACCTTGACTGT	Guerin et al., 1994
HTG4	7	9	116–137	F: CTATCTCAGTCTTCATTGCAGGAC R: CTCCCTCCCTCCCTCTGTTCTC	Ellegren et.al, 1992
HTG6	9	1	73–103	F: CCTGCTTGGAGGCTGTGATAAGAT R: GTTCACTGAATGTCAAATTCTGCT	Ellegren et.al, 1992
HTG7	5	4	114–126	F: CCTGAAGCAGAACATCCCTCCTTG R: TAAAGTGTCTGGGCAGAGCTGCT	Marklund et al., 1994
HTG10	13	21	83–105	F: CAATTCCCGCCCCACCCCCGGCA R: TTTTTATTCTGATCTGTCACATTT	Marklund et al., 1994
LEX3	12	Xq	137–160	F: ACATCTAACCAGTGCTGAGACT R: AAGGAAAAAAAGGAGGAAGAC	Coogle L.et al, 1996
VHL20	10	30	83–102	F: CAAGTCCTCTTACTTGAAGACTAG R: AACTCAGGGAGAATCTTCCTCAG	Van Haeringen et al., 1994

**Table 2 animals-10-01217-t002:** The allele spectrum of microsatellite loci in TB horses in different periods.

Locus	First Period (1990–1999)			Second Period (2000–2009)			Third Period (2010–2018)		
	Na	Ae	Ho	Na	Ae	Ho	Na	Ae	Ho
AHT4	4	3.549	0.721	4	3.498	0.716	4	3.538	0.715
AHT5	5	3.657	0.736	5	3.468	0.710	5	3.241	0.686
ASB17	6	4.116	0.764	7	4.133	0.781	7	4.031	0.763
ASB2	9	6.554	0.847	9	6.415	0.867	9	5.701	0.854
ASB23	6	4.142	0.762	6	4.505	0.770	6	4.599	0.781
CA425	7	2.114	0.524	7	2.359	0.582	7	2.288	0.547
HMS1	3	2.598	0.615	4	2.660	0.634	3	2.754	0.623
HMS2	5	2.180	0.537	5	2.025	0.508	5	1.710	0.417
HMS3	6	2.707	0.643	6	2.678	0.621	6	2.503	0.591
HMS6	5	2.257	0.561	5	2.424	0.579	5	2.542	0.604
HMS7	6	4.865	0.809	6	4.674	0.787	6	4.551	0.785
HTG10	7	4.461	0.751	7	4.496	0.746	7	4.922	0.793
HTG4	5	2.371	0.573	5	2.298	0.569	5	2.233	0.569
HTG6	6	2.612	0.614	6	2.542	0.607	6	2.556	0.623
HTG7	4	2.742	0.665	4	2.710	0.631	4	2.714	0.651
LEX3	7	4.904	0.794	7	4.667	0.748	8	4.210	0.762
VHL20	6	3.813	0.732	6	3.914	0.744	5	3.886	0.726
Average	5.71	3.508	0.685	5.82	3.498	0.682	5.77	3.411	0.676

Note: *N_a_*, number of alleles; *A_e_*, effective number of alleles; *H_e_*, expected heterozygosity.

**Table 3 animals-10-01217-t003:** The genetic parameters in the population of TB horses for 1990–2018.

Period	N	Na	Ae	Ho	He	Fis
1990–1999	1338	5.71	3.508	0.685	0.682	−0.003 ***
2000–2009	5008	5.82	3.498	0.682	0.684	0.002 ***
2010–2018	3334	5.77	3.411	0.676	0.674	−0.002 ***

Note: N, number of horses; N_a_, number of alleles; A_e_, effective number of alleles; H_o_, observed heterozygosity; H_e_, expected heterozygosity; Fis, population inbreeding level; *** *p* < 0.001.

**Table 4 animals-10-01217-t004:** The inbreeding coefficient Fx and heterozygosity of STR loci in groups of Thoroughbred horses for 1990–2018.

Group	N	Inbreeding Coefficient Fx (%)		Heterozygosity (He)
		M ± m	Lim	
**1990–1999**				
Stallions	201	0.520 ± 0.0652	0–7.23	0.677 ± 0.0133
Mares	754	0.738 ± 0.0459	0–12.50	0.687 ± 0.0038
Foals	383	0.662 ± 0.0616	0–12.50	0.685 ± 0.0054
Total	1338	0.683 ± 0.0328	0–12.50	0.685 ± 0.0017
**2000–2009**				
Stallions	833	0.758 ± 0.0427	0–13.28	0.671 ± 0.0064
Mares	2579	0.821 ± 0.0249	0–12.70	0.688 ± 0.0020
Foals	1596	0.826 ± 0.0296	0–12.70	0.678 ± 0.0025
Total	5008	0.812 ± 0.0324	0–13.28	0.682 ± 0.0016
**2010–2018**				
Stallions	651	0.903 ± 0.0496	0–9.77	0.672 ± 0.0063
Mares	1689	0.899 ± 0.0331	0–12.70	0.677 ± 0.0024
Foals	994	0.904 ± 0.0413	0–9.77	0.677 ± 0.0037
Total	3334	0.900 ± 0.0413	0–12.70	0.676 ± 0.0019

**Table 5 animals-10-01217-t005:** The distribution of TB horses by the degree of homozygosity and the level of inbreeding (Fx).

Level of Homozygosity (%)	Inbreeding Level (Fx)						
	0	0.1–1.0	1.1–2.0	2.1–3.0	3.1–4.0	>4.0	Total
0,0	20	28	17	7	3	0	75
0.1–6.25	8	12	7	0	0	3	30
6.26–12.5	171	177	75	22	12	10	467
12.6–18.75	423	432	164	60	44	31	1154
18.76–25.27	675	532	186	77	76	24	1570
25.28–31.52	791	703	310	81	106	49	2040
31.53–37.77	263	364	152	58	51	14	902
37.78–44.02	618	528	217	48	73	47	1531
44.03–50.27	466	393	170	46	50	30	1131
50.28–56.52	162	211	71	14	22	15	495
56.53–62.77	71	70	28	10	10	3	192
62.78–69.02	6	12	4	6	5	0	33
69.03–75.27	7	5	5	1	7	1	26
75.28–76.92	1	5	4	0	0	0	10
Total	3658	3472	1410	430	459	227	9680
%	37.88	35.96	14.60	4.45	4.75	2.35	100

**Table 6 animals-10-01217-t006:** The inbreeding coefficient (Fx) and homozygosity level in groups of TB horses.

Inbreeding Coefficient	Stallions		Mares		Foals	
	N	Homozygosity	N	Homozygosity	N	Homozygosity
0	638	31.69 ± 0.0184	1954	31.50 ± 0.0105	1077	32.29 ± 0.0142
0.1–1.0	630	31.23 ± 0.0185	1761	31.98 ± 0.0111	1090	32.25 ± 0.0142
1.1–2.0	251	32.71 ± 0.0296	702	31.30 ± 0.0175	458	33.38 ± 0.0220
2.1–3.0	56	28.93 ± 0.0606	225	31.22 ± 0.0309	150	31.65 ± 0.0380
3.1–4.0	72	32.99 ± 0.0554	253	34.53 ± 0.0299	136	31.82 ± 0.0399
>4.0	38	32.89 ± 0.0762	127	32.54 ± 0.0416	62	33.32 ± 0.0599
Total	1685	31.66 ± 0.0113	5022	31.81 ± 0.0066	2973	32.41 ± 0.0086

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
