# Peer review of "Dynamics of the Inbreeding Coefficient and Homozygosity in Thoroughbred Horses in Russia"

_animals, 2020, doi:10.3390/ani10071217_

Round 1

Reviewer 1 Report

I think that the subject of the work is of interest and that the topic of the manuscript is appropriate for the Journal. The information is of significant interest to the Journal's readers. However, English language should be improved and some missing information in methods should be added. Therefore, I suggest that the study could be suitable for publication pending minor revision.

Specific Comments

The title adequately reflect the major finding of the study.

The abstract adequately summarize methodology, results, and significance of the study.

The introduction section falls within the topic of the study, however, Authors should indicate the full term of acronyms which appear for the first time. Moreover, the punctuation and English language should be improved.

The section of Materials and Methods is clear for the reader and it meticulously describes the methods applied in the study. However some missing information should be added. At this regard, Authors should add a table reporting the list of microsatellite loci used according to the molecular weight: locus name, repeated sequence, primer sequence, fluorophores used, annealing temperature, Size range and number of alleles for each locus. Why authors did not perform phylogenetic analysis?

Results section as well as Discussion section is clear and well written. The findings obtained in the study were well discussed and justified with appropriate references.

The conclusion section is well written and well structured. Effectively, Authors well summarized the results and the significance of the study in this section.

Authors should check and standardize the references in the list according to journal guidelines.

Author Response

Response to Reviewer 1 Comments 

Dear reviewer! Thank You for your correct comments! 

All changes made again, as well as additions, are marked in red in the text of the article. 

If the quality of English is still not suitable, we are ready to use the MDPI service 

Point 1: The introduction section falls within the topic of the study, however, Authors should indicate the full term of acronyms which appear for the first time. Moreover, the punctuation and English language should be improved. 

Response 1: now it is L 37-40, : 

  1. Introduction

The Thoroughbred breed of horses (TB), created in England in the 17-18th centuries, has been perfected by the method of purebred breeding for many generations. The breed was formed in the result of crossing local mares with stallions of Eastern origin. The breed has three foundation stallions - Byerley Turk, Darley Arabian and Godolphin Arabian [1].  

Point 2: The section of Materials and Methods is clear for the reader and it meticulously describes the methods applied in the study. However some missing information should be added. At this regard, Authors should add a table reporting the list of microsatellite loci used according to the molecular weight: locus name, repeated sequence, primer sequence, fluorophores used, annealing temperature, Size range and number of alleles for each locus. Why authors did not perform phylogenetic analysis? 

Response 2: Table1. Information about 17 microsatellite loci used in our study  

Locus 

Number alleles 

Chrom. 

location 

Amplicon length (bp) 

Primer sequences (5’-3’) 

Reference 

AHT4 

11 

24 

140-166 

F: aaccgcctgagcaaggaagt 

R: cccagagagtttaccct 

Binns et al, 1995 

AHT5 

126-147 

F: acggacacatccctgcctgc 

R: gcaggctaagggggctcagc 

Binns et al, 1995 

ASB2 

15 

15 

237-268 

F: ccttccgtagtttaagcttctg 

R: cacaactgagttctctgatagg 

Breen et al, 1997 

ASB17 

19 

104-116 

F: gagggcggtacctttgtacc 

R: accagtcaggatctccaccg 

Breen et al, 1995 

ASB23 

14 

176-212 

F: gaggtttgtaattggaatg 

R: gagaagtcatttttaacacct 

Breen et al, 1995 

CA425 

11 

28 

224-247 

F: agctgcctcgttaattca 

R: ctcatgtccgcttgtctc 

Der Valle A. et al, 1997 

HMS1 

15 

166-178 

F: catcactcttcatgtctgcttgg 

R:ttgacataaatgcttatcctatggc 

Guerin et al, 1994 

HMS2 

12 

10 

218-238/ 

F: acggtggcaactgccaaggaag 

R: cttgcagtcgaatgtgtattaaatg 

Guerin et al, 1994 

HMS3 

10 

146-170 

F: ccaactctttgtcacataacaaga 

R: ccatcctcactttttcactttgtt 

Guerin et al, 1994 

HMS6 

154-170 

F: gaagctgccagtattcaaccattg 

R: ctccatcttgtgaagtgtaactca 

Guerin et al, 1994 

HMS7 

167-186 

F: caggaaactcatgttgataccatc 

R: gttgttgaaacataccttgactgt 

Guerin et al, 1994 

HTG4 

116-137 

F: ctatctcagtcttcattgcaggac 

R: ctccctccctccctctgttctc 

Ellegren et.al, 1992 

HTG6 

73-103 

F: cctgcttggaggctgtgataagat 

R: gttcactgaatgtcaaattctgct 

Ellegren et.al, 1992 

HTG7 

114-126 

F: cctgaagcagaacatccctccttg 

R: taaagtgtctgggcagagctgct 

Marklund et al, 1994 

HTG10 

13 

21 

83-105 

F: caattcccgccccacccccggca R:tttttattctgatctgtcacattt 

Marklund et al, 1994 

LEX3 

12 

Xq 

137-160 

F: acatctaaccagtgctgagact R: aaggaaaaaaaggaggaagac 

Coogle L.et al, 1996 

VHL20 

10 

30 

83-102 

F: caagtcctcttacttgaagactag R: aactcagggagaatcttcctcag 

Van Haeringen et al, 1994 

Point 3:Authors should check and standardize the references in the list according to journal guidelines.

Response 3: the citation list is reformatted: 

Edwards, E.H. The ultimate horse book; Dorling Kindersley Ld: London, GB, 1991, pp. 34-35. 

Budyonny, C. M. The Thoroughbred horse. In The book about the horse, Vol. 1; Selkhozgiz: Moskow, USSR, 1952, pp. 383-412. 

Barmintsev, Y. N. Horse breeding and equestrian sport the USSR; Kolos: Moscow, USSR, 1972, pp. 41-43. 

Cunningham, E.P.; Dooley, J.J.; Splan, R.K.; Bradley, D.G. Microsatellite diversity, pedigree relatedness and the contributions of founder lineages to thoroughbred horses. Animal Genetics 2001, 32(50), 360-364.  

Binns, M.M.; Boehler, D.A.; Bailey, E.; Lear, T.L.; Cardwell, G.M.; Lambert, D.H. Inbreeding in the Thoroughbred horse. Animal Genetics 2012, 43, 3, 340-342; DOI: 10.1111/j.1365-2052.2011.02259.x. 

Schaefer, R.J.; Schubert, M.; Bailey, E.; Bannasch, D.L.; Barrey, E.; Bar-Gal, G.K. et al.  Developing a 670k genotyping array to tag ~2M SNPs across 24 horse breed. BMC Genomics 2017, 18, 565 DOI: 10.1186/s12864-017-3943-8. 

Kuznetsov, V. M. Inbreeding in livestock: methods of assessment and forecast; NIISH of the North-East: Kirov, Russia, 2000, P. 33-43 

Nicholas, F.M. Introduction to veterinary genetics. 2nd Ed.; Blackwell Publishing: Oxford University Press, 2008; P.183-189. 

Pern, E.M. The role of inbreeding in the improvement of horse and trotter breeds of horses. In The use of inbreeding in animal husbandry; Nauka: Moscow, USSR, 1977, pp. 46-52. 

FAO Molecular tool for exploring genetic diversity. The Second Report on the State of the World’s Animal Genetic Recourses for Food and Agriculture Organization of the United Nations. Rome, 2015, 431-449..ed. Scherf B.D.; Pilling. D.; FAO Commission on Genetic Resources for Food andAgriculture Assessments. Rome (available at http://www.fao.org/3/a-i4787e/index.html) 2015 

Canon, J.; Checa, M.L.; Carleos, C.; Vega-Pla, J.L.; Vallejo, M.; Dunner, S. The genetic structure of Spanish Celtic horse breeds inferred from microsatrllite data. Animal Genetics 2000, 31 (1), 39-48. DOI: 10.1046 / j.1365-2052.2000.00591.x  

Kalashnikov, V.V.; Khrabrova, L.A.; Zaitcev, A.M.; Zaitceva, M.A.; Kalinkova, L.V. Polymorphism of microsatellite DNA in horses of stud and local breeds. Agricultural Biology 2011, 2, 41-45.  

Ling, Y.H.; Guan, W.J.; Cheng, Y.J.; Wang, Y.P.; Han, J.L.; Mang, l.; Zhao, Q.J.; He, X.H.; Pu, Y.B.; Fu B.L. Evaluation of the genetic diversity and population structure of Chinese indigenous horse breeds using 27 microsatellite loci. Animal Genetics 2011, 42(1), 56-63. DOI: 10.1111 / j.1365-2052.2010.02067.x 

Putnova, L.; Stohl, R.; Vrtkova I. Genetic monitoring of horses in the Czech Republic: A large scale study with a focus on the Czech autochthonous breeds. Journal of Animal Breeding and Genetics 2018, 135(1), 73-83. DOI: 10.1111/jbg.12313.  

Cosenza, M.; La Rosa, V.; Rosati, R.; Chiofalo V. Genetic diversity of the Italian thoroughbred horse population. Italian Journal of Animal Science 2019, 1, 538-545. DOI: 10.1080/1828051X.2018.1547128. 

Curik, I.; Zechner, P.; Solkner, J.; Achmann, R.; Bodo, I.; Dovc, P.; Kavar, T.; Marti, E.; Brem, G. Inbreeding, microsatellite heterozygosity and morphological traits in Lipizzan horses. The Journal of Heredity 2003, 94(2), 125-132. DOI: 10.1093/jhered/esg029 

Khrabrova, L.A.; Blochina, N.V.; Ustiantseva, A.V. Inbreeding and the level of homozygosis of microsatellite loci in horses (Equus caballus) of Orlov Trotter breed .Agricultural Biology 2014, 49, 35-41. DOI: 10.15389/agrobiology.2014.4.35rus 

Langlois, B. A review on the methods of parentage and inbreeding analysis with molecular markers. Conservation genetics of endangered horse breeds, EAAP publication , 6 september 2004 116, Bled, Slovenia, 2005; 35-54. 

Pirault, P.; Danvy, S.; Verrier, E.; Leroy, G. Genetic structure and gene flows within horses: A genealogical study at the Franch population scale. Plos one 2013, 8(4), 615448. DOI: 10.1371/jornal.pone.0061544. 

Khrabrova, L.A. Monitoring of the genetic structure of breeds in horse breeding. Russian Agricultural Sciences 2008, 34, 4, 261-263; DOI: 10.3103/S1068367408040150. 

Khrabrova, L.A.; Blohina, N.V.; Suleymanov, O.I.; Rozhdestvenskaya, G.A.; Pustovoy, V.F. Assessment of line differentiation in the Thoroughbred horsebreed using DNA microsatellite loci. Vavilov Journal of. Genetics and Breeding. 2019, .23, 569-574. DOI: 10.18699/VJ19.526 

Van de Goor, L.H.P.; Panneman, H.; Van Haeringen, W.A. A proposal for standardization in forensic equine DNA typing: allele nomenclature for 17 equine-specific STR loci. Animal Genetics 2010, 41 (2), 122-127. DOI: 10.1111/j.1365-2052.2009.01975.x 

Weir, B.S. Genetic data analysis II. Sinauer Associates: Sunderland, MA. 1996. 

Rukavina D., Hasanbasic D., Ramic J., Zahirovic A., Ajanovic A., Beganovic K., Durnic-Pasic A., Kalamujic B., Pojskic N. Genetic diversity of Thoroughbred horse population from Bosnia and  Herzegovina based on 17 microsatellite markers. The Japanese Journal of Veterinary Research 2016, 64(3), 215-220. 

Jungwoo, E.; Jeong-An, G.; Bong-Hwan, C.; Kfoals-Tag, D.; Byung-Wook, C.; Heui-Soo K. Genetic profiling of Thoroughbred racehorses by microsatellite marker analysis. Genes and Genomics 2014. 36, 119-123. 

Shelyov A.V.; Melnyk O.V.; Suprun I.O.; Spyrydonov S.V.; Melnychuk S.D.; Dzitsiuk V.V.; Gorka B.M. The comparative analysis of the allele pool of Thoroughbred horses in different countries. Iranian Journal of Applied Animal Science 2014, 4(3), 637-641. 

Cosenza, M.; La Rosa,V.; Rosati, R.; Chiofalo, V. Genetic diversity of the Italian thoroughbred population. Italian Journal of Applied Animal Science 2019, 1-8. DOI: 10.1080/1828051x.2018.1547128. 

Khrabrova, L.A.; Blohina, N.V. Genetic monitoring of the Thoroughbred breed on loci of DNA microsatellite. Journal of Genetics and breeding animals 2018, 3, 11-16. DOI: 10.31043/2410-2733-2018-3-11-16. 

Levontin, R. C.The genetic basis of evolutionary change. Mir: Moscow, USSR, 1978, p.351. 

Altukhov, Y.P. Genetic processes in populations. Mir: Moscow, Russia, 2003, p.247  

Blochina N.V; Khrabrova L.A. Characteristics of Thoroughbred stallions of different lines by microsatellite loci. Genetics and Selection Animal 2019, 3, 11-17 DOI: 10.31043/2410-2733-2019-3-11-17 

Todd, E.T.; Ho, S.W.; Thomson, P.C., And R.A.; Velie, B.D., Hamiton, N.a. Founder-specific inbreeding depression affects racing performance in Thoroughbred horses. Scientific Reports  2018. 8: 6167. 

Thank you very much for your review! 

Reviewer 2 Report

Line 32: TB should be defined in body not abstract.

L32-33 need citation.

L35 needs a citation.

L43: I disagree inbreeding is an indirect consequence of targeted selection through artificial selection this sentence needs to be rewritten.

Paragraph at L43 stakes a rather odd stance on the importance of reducing heterozygosity when most breeders are trying strike a balance between purifying selection and the inbreeding of deleterious mutations. I don’t believe that controlling homozygosity is the goal of most breeding programs so much as removal of deleterious alleles. This needs better framing.

Citation formatting needs significant improvement.

L55 sentence is poorly worded

L54 should be formatted to 20th century not roman numerals.

L66 should be around the world.

L66 is a long run on grammar needs correction

L71 duplicate period.

L120 references something that cannot be seen, no significant results were demonstrated.

Entire paragraph of L120 needs to be written to better display results. Inference is being made to results that are not demonstrated.

Table 1 and 2

Subscript needs to be done for table. Tables need significant reformatting and legends better defined.

Roman numerals needs to be removed.

Table 4 , needs to be changed to . for traditional readers.

L211 starts with several citations when it is stating results from the current study. Is that purposeful? This appears to be an error.

L215 SNP is not defined.

Discussion is significantly lacking. It fails to appropriately interpret the results presented in the study. Significant expansion on the results needs to be done in order to properly highlight the numerous results of the study. Little discussion is made in regards to the low degree of homozygosity of the TB groups.

Formatting for the journal has not been properly done.

Author Response

Response to Reviewer 2 Comments 

Dear reviewer! Thank You for your correct comments! 

All changes made again are marked in red in the text of the article. 

If the quality of English is still not suitable, we are ready to use the MDPI service 

Point 1: L32-33 need citation. 

L35 needs a citation. 

Response 1: now it is L 37-40, citation indicated: 

  1. Introduction

The Thoroughbred breed of horses (TB), created in England in the 17-18th centuries, has been perfected by the method of purebred breeding for many generations. The breed was formed in the result of crossing local mares with stallions of Eastern origin. The breed has three foundation stallions - Byerley Turk, Darley Arabian and Godolphin Arabian [1].  

Point 2: L43: I disagree inbreeding is an indirect consequence of targeted selection through artificial selection this sentence needs to be rewritten. 

Paragraph at L43 stakes a rather odd stance on the importance of reducing heterozygosity when most breeders are trying strike a balance between purifying selection and the inbreeding of deleterious mutations. I don’t believe that controlling homozygosity is the goal of most breeding programs so much as removal of deleterious alleles. This needs better framing. 

Response 2: now it is L 45-50:  Long-term intensive selection of horses for racing performance and wide use of the best stallions in breeding inevitably led to the accumulation of well-known names in their pedigrees. According to Cunningham et al [4] TB horses have narrow genetic base, 78% of alleles in the current population are from 30 progenitors, in whole one stallion Godolphin Arabian is responsible for 95% of paternal lines. This may lead to an increase of inbreeding level and homozygosity rate of genomes of TB horses [5, 6]. 

Point 3:Citation formatting needs significant improvement. 

Response 3: the citation list is reformatted: 

Edwards, E.H. The ultimate horse book; Dorling Kindersley Ld: London, GB, 1991, pp. 34-35. 

Budyonny, C. M. The Thoroughbred horse. In The book about the horse, Vol. 1; Selkhozgiz: Moskow, USSR, 1952, pp. 383-412. 

Barmintsev, Y. N. Horse breeding and equestrian sport the USSR; Kolos: Moscow, USSR, 1972, pp. 41-43. 

Cunningham, E.P.; Dooley, J.J.; Splan, R.K.; Bradley, D.G. Microsatellite diversity, pedigree relatedness and the contributions of founder lineages to thoroughbred horses. Animal Genetics 2001, 32(50), 360-364.  

Binns, M.M.; Boehler, D.A.; Bailey, E.; Lear, T.L.; Cardwell, G.M.; Lambert, D.H. Inbreeding in the Thoroughbred horse. Animal Genetics 2012, 43, 3, 340-342; DOI: 10.1111/j.1365-2052.2011.02259.x. 

Schaefer, R.J.; Schubert, M.; Bailey, E.; Bannasch, D.L.; Barrey, E.; Bar-Gal, G.K. et al.  Developing a 670k genotyping array to tag ~2M SNPs across 24 horse breed. BMC Genomics 2017, 18, 565 DOI: 10.1186/s12864-017-3943-8. 

Kuznetsov, V. M. Inbreeding in livestock: methods of assessment and forecast; NIISH of the North-East: Kirov, Russia, 2000, P. 33-43 

Nicholas, F.M. Introduction to veterinary genetics. 2nd Ed.; Blackwell Publishing: Oxford University Press, 2008; P.183-189. 

Pern, E.M. The role of inbreeding in the improvement of horse and trotter breeds of horses. In The use of inbreeding in animal husbandry; Nauka: Moscow, USSR, 1977, pp. 46-52. 

FAO Molecular tool for exploring genetic diversity. The Second Report on the State of the World’s Animal Genetic Recourses for Food and Agriculture Organization of the United Nations. Rome, 2015, 431-449..ed. Scherf B.D.; Pilling. D.; FAO Commission on Genetic Resources for Food andAgriculture Assessments. Rome (available at http://www.fao.org/3/a-i4787e/index.html) 2015 

Canon, J.; Checa, M.L.; Carleos, C.; Vega-Pla, J.L.; Vallejo, M.; Dunner, S. The genetic structure of Spanish Celtic horse breeds inferred from microsatrllite data. Animal Genetics 2000, 31 (1), 39-48. DOI: 10.1046 / j.1365-2052.2000.00591.x  

Kalashnikov, V.V.; Khrabrova, L.A.; Zaitcev, A.M.; Zaitceva, M.A.; Kalinkova, L.V. Polymorphism of microsatellite DNA in horses of stud and local breeds. Agricultural Biology 2011, 2, 41-45.  

Ling, Y.H.; Guan, W.J.; Cheng, Y.J.; Wang, Y.P.; Han, J.L.; Mang, l.; Zhao, Q.J.; He, X.H.; Pu, Y.B.; Fu B.L. Evaluation of the genetic diversity and population structure of Chinese indigenous horse breeds using 27 microsatellite loci. Animal Genetics 2011, 42(1), 56-63. DOI: 10.1111 / j.1365-2052.2010.02067.x 

Putnova, L.; Stohl, R.; Vrtkova I. Genetic monitoring of horses in the Czech Republic: A large scale study with a focus on the Czech autochthonous breeds. Journal of Animal Breeding and Genetics 2018, 135(1), 73-83. DOI: 10.1111/jbg.12313.  

Cosenza, M.; La Rosa, V.; Rosati, R.; Chiofalo V. Genetic diversity of the Italian thoroughbred horse population. Italian Journal of Animal Science 2019, 1, 538-545. DOI: 10.1080/1828051X.2018.1547128. 

Curik, I.; Zechner, P.; Solkner, J.; Achmann, R.; Bodo, I.; Dovc, P.; Kavar, T.; Marti, E.; Brem, G. Inbreeding, microsatellite heterozygosity and morphological traits in Lipizzan horses. The Journal of Heredity 2003, 94(2), 125-132. DOI: 10.1093/jhered/esg029 

Khrabrova, L.A.; Blochina, N.V.; Ustiantseva, A.V. Inbreeding and the level of homozygosis of microsatellite loci in horses (Equus caballus) of Orlov Trotter breed .Agricultural Biology 2014, 49, 35-41. DOI: 10.15389/agrobiology.2014.4.35rus 

Langlois, B. A review on the methods of parentage and inbreeding analysis with molecular markers. Conservation genetics of endangered horse breeds, EAAP publication , 6 september 2004 116, Bled, Slovenia, 2005; 35-54. 

Pirault, P.; Danvy, S.; Verrier, E.; Leroy, G. Genetic structure and gene flows within horses: A genealogical study at the Franch population scale. Plos one 2013, 8(4), 615448. DOI: 10.1371/jornal.pone.0061544. 

Khrabrova, L.A. Monitoring of the genetic structure of breeds in horse breeding. Russian Agricultural Sciences 2008, 34, 4, 261-263; DOI: 10.3103/S1068367408040150. 

Khrabrova, L.A.; Blohina, N.V.; Suleymanov, O.I.; Rozhdestvenskaya, G.A.; Pustovoy, V.F. Assessment of line differentiation in the Thoroughbred horsebreed using DNA microsatellite loci. Vavilov Journal of. Genetics and Breeding. 2019, .23, 569-574. DOI: 10.18699/VJ19.526 

Van de Goor, L.H.P.; Panneman, H.; Van Haeringen, W.A. A proposal for standardization in forensic equine DNA typing: allele nomenclature for 17 equine-specific STR loci. Animal Genetics 2010, 41 (2), 122-127. DOI: 10.1111/j.1365-2052.2009.01975.x 

Weir, B.S. Genetic data analysis II. Sinauer Associates: Sunderland, MA. 1996. 

Rukavina D., Hasanbasic D., Ramic J., Zahirovic A., Ajanovic A., Beganovic K., Durnic-Pasic A., Kalamujic B., Pojskic N. Genetic diversity of Thoroughbred horse population from Bosnia and  Herzegovina based on 17 microsatellite markers. The Japanese Journal of Veterinary Research 2016, 64(3), 215-220. 

Jungwoo, E.; Jeong-An, G.; Bong-Hwan, C.; Kfoals-Tag, D.; Byung-Wook, C.; Heui-Soo K. Genetic profiling of Thoroughbred racehorses by microsatellite marker analysis. Genes and Genomics 2014. 36, 119-123. 

Shelyov A.V.; Melnyk O.V.; Suprun I.O.; Spyrydonov S.V.; Melnychuk S.D.; Dzitsiuk V.V.; Gorka B.M. The comparative analysis of the allele pool of Thoroughbred horses in different countries. Iranian Journal of Applied Animal Science 2014, 4(3), 637-641. 

Cosenza, M.; La Rosa,V.; Rosati, R.; Chiofalo, V. Genetic diversity of the Italian thoroughbred population. Italian Journal of Applied Animal Science 2019, 1-8. DOI: 10.1080/1828051x.2018.1547128. 

Khrabrova, L.A.; Blohina, N.V. Genetic monitoring of the Thoroughbred breed on loci of DNA microsatellite. Journal of Genetics and breeding animals 2018, 3, 11-16. DOI: 10.31043/2410-2733-2018-3-11-16. 

Levontin, R. C.The genetic basis of evolutionary change. Mir: Moscow, USSR, 1978, p.351. 

Altukhov, Y.P. Genetic processes in populations. Mir: Moscow, Russia, 2003, p.247  

Blochina N.V; Khrabrova L.A. Characteristics of Thoroughbred stallions of different lines by microsatellite loci. Genetics and Selection Animal 2019, 3, 11-17 DOI: 10.31043/2410-2733-2019-3-11-17 

Todd, E.T.; Ho, S.W.; Thomson, P.C., And R.A.; Velie, B.D., Hamiton, N.a. Founder-specific inbreeding depression affects racing performance in Thoroughbred horses. Scientific Reports  2018. 8: 6167. 

Point 4:L55 sentence is poorly worded 

L54 should be formatted to 20th century not roman numerals. 

L66 should be around the world. 

L66 is a long run on grammar needs correction 

Response 4: now it is L 61-72:  

The actual rating of homozygosity level at the individual and population levels became possible only at the end of the 20th century thanks to the development of methods for studying gene and DNA polymorphism. As a result of the widespread use of microsatellite loci in the control of animal origin this type of highly polymorphic genetic markers has been successfully used to study the genetic features of breeds and their phylogenetic relationships [11, 12, 13, 14]. Characteristics of genetic parameters of populations, among which one of the most important is heterozygosity level, allows us to assess the impact of the breeding system on the the level of inbreeding [16, 17, 18, 19]. 

Assessment of inbreeding and homozygosity levels in TB horses is especially relevant as the breed has been improved only by the method of purebred breeding for more than three centuries. Monitoring of the state of genetic diversity in the Russian population of TB horses in the period from 1980 to 2005 revealed a tendency of a certain decrease in the level of polymorphism and the rate of heterozygosity, especially in the group of stallions [20]. 

Point 5:L71 duplicate period. 

Response 5: now it is L 77-78:  

The purpose of our research was to assess the genetic diversity and the level of inbreeding in the TB breed in Russia over the period 1990-2018. 

Point 6:L120 references something that cannot be seen, no significant results were demonstrated. 

Entire paragraph of L120 needs to be written to better display results. Inference is being made to results that are not demonstrated. 

Response 6: now it is L 127: 

As a result of genotyping of TB horses, 100 alleles were identified by 17 panel STR loci, among which there were the most typical 70 variants, the frequency of which exceeded 0.05. In the periods under study the number of rare alleles in STR loci in tested horses varied slightly in the range from 27 to 29 and was highest in the second period (2000-2009), which is associated with intensive import of TB horses from Europe and America. During this period the allele pool of the domestic population was supplemented with three new rare alleles ASB17H, HMS1L and LEX3I due to the genotypes of imported mares. But in the next generation, in horses born in 2010-2018, the HMS1L allele was lost. The total number of alleles on 17 STR loci is only 98 in the modern Russian population of TB horses (3-rd period).  The breed maintains its genetic structure stably despite continued imports of horses from other countries.  

Point 7:Table 1 and 2 

Subscript needs to be done for table. Tables need significant reformatting and legends better defined. 

Roman numerals needs to be removed. 

Table 4 , needs to be changed to . for traditional readers. 

Response 7: tables are reformatted in accordance with the comments, and new ones have also been added, Roman numerals are replaced 

Point 8:L211 starts with several citations when it is stating results from the current study. Is that purposeful? This appears to be an error. 

L215 SNP is not defined. 

Discussion is significantly lacking. It fails to appropriately interpret the results presented in the study. Significant expansion on the results needs to be done in order to properly highlight the numerous results of the study. Little discussion is made in regards to the low degree of homozygosity of the TB groups. 

Response 8:  

L 222: 

The results of genotyping of 9680 Thoroughbred horses, registered in Russian Stud Book , show that the studied group has typical spectrum of alleles of microsatellite loci for the breed and is characterized by a high degree of consolidation. Only insignificant differences in frequency of rare alleles were revealed with TB populations of foreign countries. For example, a small population of Thoroughbred horses from Bosnia and Herzegovina [24] had higher variability of the same STR loci (in total 103 alleles). The results of comparative analysis of genetic structure of different TB populations [25, 26, 27] show that the TB horses have their own pool of alleles that remain stable for generations. The modern Russian TB population is characterized by a fairly high level of genetic diversity (Ae=3, 41, Ho=0,676) and lack of population`s inbreeding (Fst= -0,002). 

In spite of expectations, periodic import of TB horses from the United States and European countries had no significant impact on the population parameters of the genetic structure of TB horses in Russia. Moreover, it led to an increase in inbreeding coefficient and level of homozigosity. Comparative analysis of genetic diversity in groups of horses born in Russia and in imported horses showed that domestic stallions and mares had higher heterozygosity level [28], which was confirmed by the genetic parameters of the breed in the period 1990-2000. 

The history of TB horse breeding in a closed Stud Book for three centuries could result in reduction  of genetic diversity of the TB horses. Indeed, genomic analysis using the 670k genotyping arrey showed that the TB breed has the highest level of homozygosity [6]. The results of genome-wide analyses of TB horses using SNP50 bead chip indicate that inbreeding in the breed has increased over the past 45 years with r = 0,24, P<0,001 [5]. The trend of a certain increase in the level of inbreeding and the degree of homozygosity of the TB horses in Russia generally reflects the global trend of decreasing genetic diversity in this breed.  

It should be noted that the studied population steady maintained a certain level of heterozygosity (Ho = 0.685-0.676), both during the period of partial isolation (1990-1999) and during the intensive import of horses (2000-2009). Obviously, this can serve as a proof of the hypothesis of the existence of optimal heterozygosity of populations [29], as well as maintaining a stable ratio of the intra- and interpopulation components of genetic diversity in the population system [30]. Due to its large number and breeding in various countries, the breed retains a significant reserve of variability, which is also confirmed by the genetic differences in its linear structure [31]. 

Obviously, the inbreeding coefficient at the pedigree level does not quite adequately reflect the degree of homozygosity of an individual [7]; the much more accurate method is the genomic assessment using of a kit of SNP [18, 32]. In our studies, the level of homozygosity of outbred horses by 17 standard loci varied in a wide range from 0 to 76%, and the absence of duplicate ancestors in the pedigree of a horse cannot guarantee its high heterozygosity. Similar results were obtained in the analysis of relation of the degree of homozygosity microsatellite loci and the level of inbreeding in 1194 Orlov Trotter horses [17]. It showed that the degree of homozygosity of the outbred Orlov Trotter varies within a wide range from 0 up to 75% and the breed also had low coefficient of correlation between the degree of homozygosity of microsatellite loci and the inbreeding level (0,079). 

It`s obvious that the Wright’s inbreeding coefficient reflects the degree of homozygosity of animals very approximately, therefore genome monitoring is very important for further improvement of the Thoroughbred breed. 

Thank you very much for your review! 

Round 2

Reviewer 2 Report

The edits addressed many of my previous concerns on the manuscript. There still needs extensive rewrites to fix language and style formats appropriate for this journal. The use of ( , ) instead of ( .  ) for numerical decimals needs to be addressed throughout the entire manuscript. Table 3 N needs to either be capitalized or lower case and standardized throughout the paper.  

Overall this paper presents a fine assessment of genetic diversity in TB horses. 

Author Response

Dear reviewer! Thank you again for your correct comments!

All newly made changes and additions are marked in color in the text of the article (the manuscript is attached below)

Point 1: The edits addressed many of my previous concerns on the manuscript. There still needs extensive rewrites to fix language and style formats appropriate for this journal. The use of ( , ) instead of ( .  ) for numerical decimals needs to be addressed throughout the entire manuscript. Table 3 N needs to either be capitalized or lower case and standardized throughout the paper. 

Response 1: We used the MDPI service to correct errors in English, all changes are noted in the manuscript. All commas in numbers are corrected to dots, N is standardized in all tables.
